# Catalytic Conversion of Glucose into Levulinic Acid Using 2-Phenyl-2-Imidazoline Based Ionic Liquid Catalyst

**DOI:** 10.3390/molecules26020348

**Published:** 2021-01-12

**Authors:** Komal Kumar, Mukesh Kumar, Sreedevi Upadhyayula

**Affiliations:** Department of Chemical Engineering Indian Institute of Technology Delhi, New Delhi, Hauz Khas, New Delhi 110016, India; komalkumar924@gmail.com (K.K.); mukeshbahl924@gmail.com (M.K.)

**Keywords:** hydrothermal conversion, ionic liquid catalyst, biomass, levulinic acid

## Abstract

Levulinic acid (LA) is an industrially important product that can be catalytically valorized into important value-added chemicals. In this study, hydrothermal conversion of glucose into levulinic acid was attempted using Brønsted acidic ionic liquid catalyst synthesized using 2-phenyl-2-imidazoline, and 2-phenyl-2-imidazoline-based ionic liquid catalyst used in this study was synthesized in the laboratory using different anions (NO_3_, H_2_PO_4_, and Cl) and characterized using ^1^H NMR, TGA, and FT-IR spectroscopic techniques. The activity trend of the Brønsted acidic ionic liquid catalysts synthesized in the laboratory was found in the following order: [C_4_SO_3_HPhim][Cl] > [C_4_SO_3_HPhim][NO_3_] > [C_4_SO_3_HPhim][H_2_PO_4_]. A maximum 63% yield of the levulinic acid was obtained with 98% glucose conversion at 180 °C and 3 h reaction time using [C_4_SO_3_HPhim][Cl] ionic liquid catalyst. The effect of different reaction conditions such as reaction time, temperature, ionic liquid catalyst structures, catalyst amount, and solvents on the LA yield were investigated. Reusability of [C_4_SO_3_HPhim][Cl] catalyst up to four cycles was observed. This study demonstrates the potential of the 2-phenyl-2-imidazoline-based ionic liquid for the conversion of glucose into the important platform chemical levulinic acid.

## 1. Introduction

Fossil fuel resources such as crude-oil, coal, and natural gas are currently used for liquid fuels and fulfill the demand for energy for the majority of the world population [1,2]. Growing transportation and industrial needs of both the developed and developing nations has created challenges of greenhouse gases emissions and depletion of fossil fuel resources [3]. Hence, there is a need to explore clean and renewable energy alternative resources that can substitute these fossil fuels [4,5]. Lignocellulosic biomass and its derivatives have the potential to become the best alternative liquid transportation fuel due to their energy densities and hydrocarbon fuel compatibility with internal combustion engines [6,7]. Lignocellulosic biomass derived from non-food crops and agricultural waste is a renewable and clean alternative energy source that can also be catalytic converted into different types of commodity chemicals that can be used as fossil-fuel alternatives [8,9].

Catalytic conversion of lignocellulosic biomass to value-added products such as 5-hydroxymethyl furfural (5-HMF) [10], levulinic acid (LA) [11], acetic acid (AA), formic acid (FA), and lactic acid [12] is an appealing way to produce plastics, paper, pharmaceuticals, and transportation fuels [13,14]. Among all these, researchers have identified biomass-derived LA and 5-HMF as highly valuable molecules, since they can be employed as intermediate chemicals in the production of a broad range of industrially important chemicals and liquid transportation fuels components such as 2,5-dimethylfuran (DMF) [15], 2,5-bis(hydroxymethyl)-furan (BHMF), HMF-acetal [16,17], and 2-hydroxymethyl-5-methylfuran (HMMF). Similarly, the valorization of LA can generate clean liquid fuel precursors such as γ-valerolactone and its ether and ester products [18,19]. Hence, the conversion of lignocellulosic biomass and its derivatives into LA is more desirable and has attracted great attention in recent years. Significant efforts have been made to convert lignocellulosic biomass and its derivatives, such as glucose and fructose, into LA using acid catalysis [20,21]. Many researchers have investigated the synthesis of LA in homogeneous phases using mineral acid catalysts, such as H_2_SO_4_, H_3_PO_4_ [22], HCl [23], and organic acids; and heterogeneous solid acid catalysts, such as mesoporous SBA-15 and polymeric nanotubes [2,24,25]. In the case of homogeneous acid catalysts, moderate to high yields of LA have been achieved via the catalytic conversion of biomass and its derived feedstock. However, the use of the mineral acid catalyst in the production of LA poses some challenges such as separation, purification of LA, handling of the homogeneous acid catalyst and their high toxicity, and the generation of the acidic waste [25,26,27].

The above addressed issues can be avoided by using a solid acid catalyst in the catalytic synthesis of LA, but many of these heterogeneous catalysts give low selective yield of the desired products and require higher catalyst/substrate ratios, high reaction temperatures, and long reaction times [28,29]. For instance, Haonan Qu and co-workers used a solid acid catalyst (metal–organic framework) containing Brønsted and Lewis acidity and reported a 57.9% yield of LA at 150 °C within a 9 h reaction time [24]. Similarly, Vijay Bhooshan Kumar and co-workers used Ga salt of molybdophosphoric acid (GaHPMo) for glucose conversion to LA and reported a 56% yield at 150 °C and 10 h reaction time [30]. Wei Zeng and co-workers used MFI-Type Zeolite (ZRP) zeolite for LA production and reported a 35% yield from glucose conversion at 180 °C and an 8 h reaction time [31]. Therefore, alternate promising new catalytic systems are needed. In this context, ionic liquids (ILs) are a good choice that can be used as catalysts and solvents for biomass conversion into value-added products and intermediates like LA [32].

Acidic ILs have received considerable attention in many fields, due to their unique characteristics, and have a high potential to replace the conventional mineral and solid acid acidic catalysts. Brønsted acidic ILs are recognized as recyclable, tunable acid catalysts [33] that can be used as solvents and catalysts [34]. Their low vapor pressure at room temperature makes them highly attractive for use in low temperature reactions [35]. Different types of IL catalysts have been reported for biomass conversion into total reducing sugars [36], 5-HMF [37], LA [28], and other compounds [38,39]. Our group has also synthesized and employed multifunctional IL catalysts that have –SO_3_H, COOH, and OH functional groups in the direct conversion of glucose to LA under hydrothermal conditions [21].

This work aims at exploring the catalytic activity of the 2-phenyl-2-imidazoline-based IL in the glucose conversion to LA. Synthesis of this catalyst is followed by characterization using ^1^H, ^13^C NMR, TGA, and FT-IR spectroscopic techniques. The catalytic activity of the synthesized catalyst was evaluated in the glucose conversion to LA in a batch autoclave. Kinetic investigations in the batch reactor led to optimization of reaction conditions, time and temperature, catalyst amount, and solvent effect based on catalytic performance.

## 2. Results and Discussion

### 2.1. Characterization of the Synthesized Brønsted Acidic Ionic Liquid Acid Catalyst

#### 2.1.1. Characterization of the Catalyst by ^1^H NMR

The laboratory-synthesized Brønsted acidic ionic liquid catalyst was characterized using the ^1^H NMR spectrum as follows: (1-butyl sulfonic acid-2-phenylimidazoline chloride) [C_4_SO_3_HPhim][Cl] ^1^H NMR: (400 MHz, D_2_O): δ (ppm) 1.48 (m, 2H), 1.58 (m, 2H), 2.60 (t, 2H), 3.15 (t, 2H), 3.96 (d, 4H), 7.60 (m, 5H). (1-butyl sulfonic acid-2-phenylimidazoline nitrate) [C_4_SO_3_HPhim][NO_3_] ^1^H NMR: (400 MHz, D_2_O): δ (ppm) 1.40 (m, 2H), 1.50 (m, 2H), 2.53 (t, 2H), 3.10 (t, 2H), 3.90 (d, 4H), 7.50 (m, 5H). (1-butyl sulfonic acid-2-phenylimidazoline hydrogen phosphate) [C_4_SO_3_HPhim][H_2_PO_4_] ^1^H NMR: (400 MHz, D_2_O): δ (ppm) 1.2 (m, 2H), 1.35 (m, 2H), 2.50 (t, 2H), 3.10 (t, 2H), 3.80 (d, 4H), 7.40 (m, 5H).

#### 2.1.2. Characterization of the Catalyst by FT-IR

The characterization of the laboratory-synthesized [C_4_SO_3_HPhim][Cl], [C_4_SO_3_HPhim][NO_3_], and [C_4_SO_3_HPhim][H_2_PO_4_] IL catalysts was done using FT-IR, and the spectra are shown in Figure 1a–c. The [C_4_SO_3_HPhim][Cl] FT-IR spectrum displayed a broad peak at 3300–3600 cm^−1^, which represents O–H stretching of the sulfonic acid (–SO_3_H) functional group in the catalyst. A peak appeared in the range of 2810–3020 cm^−1^ which may be attributed to the C–H stretching vibrations of the aliphatic butyl chain attached to the imidazoline ring [40]. Appearance of a strong peak in the range of 1580–1606 cm^−1^ was attributed to the C=C stretching of the benzene ring, which was associated with a five membered ring [41]. In the case of the [C_4_SO_3_HPhim][NO_3_] ionic liquid catalyst, a sharp peak appeared at 1335 cm^−1^ attributed to the (N‒O) symmetric stretching [42]. In the case of the IL [C_4_SO_3_HPhim][H_2_PO_4_], a broad band that appeared at 2550–2700 cm^−1^ confirmed the existence of the P–O–H bond within the IL [43]. A sharp peak in the range of 1115–1166 cm^−1^ was associated with the C–N bond which was present in the five membered ring of the IL acid catalyst [44]. A peak at 1030–1037 cm^−1^ was associated with the S=O bond in SO_3_H [45].

#### 2.1.3. Thermal Stability of the IL Catalyst ([C_4_SO_3_HPhim][Cl])

Thermal stability of the laboratory-synthesized Brønsted acidic IL [C_4_SO_3_HPhim][Cl] catalyst was studied by the thermogravimetric analysis (TGA) method under a nitrogen atmosphere over a temperature range from 30 to 550 °C, and the obtained results are shown in Figure 2. From Figure 2, it can be observed that the 1st onset of weight loss in the ionic liquid catalyst [C_4_SO_3_HPhim][Cl] was observed at around 175 °C, and overall, 5% weight loss was observed. From the TGA analysis it is clear that the synthesized IL catalyst showed good thermal stability up to 200 °C, and weight loss above 200 °C was evident from the change of the slope of TGA and showed high weight loss around 398–400 °C. The TGA confirmed that the laboratory-synthesized catalyst had high thermal stability in the range of our catalytic reaction temperatures.

#### 2.1.4. Hammett Acidity of the Synthesized Ionic Liquids

Brønsted acidic strength of the laboratory-synthesized IL catalysts was determined using UV–vis spectroscopy. The Hammett function values were calculated using Equation (1) and are reported in Table 1. The results represented in Table 1 show that the Brønsted acidic IL catalyst [C_4_SO_3_HPhim][Cl] had the lowest $H_{0}$ value (2.17), whereas the catalyst [C_4_SO_3_HPhim][H_2_PO_4_] had the highest $H_{0}$ value (3.00). Lower $H_{0}$ values of [C_4_SO_3_HPhim][Cl] signified high acidic strength of this IL catalyst. On the basis of $H_{0}$ values, the order of the acidity of the IL catalysts was found to be as follows: [C_4_SO_3_HPhim][Cl] > [C_4_SO_3_HPhim][NO_3_] > [C_4_SO_3_HPhim][H_2_PO_4_].

### 2.2. Catalyst Activity Testing

#### 2.2.1. Screening of the Ionic Liquid Catalyst

Figure 3 shows the activity of the IL catalysts that have three different anions (Cl, NO_3_, and H_2_PO_4_) in the glucose conversion to LA at 180 °C for 3 h. It is clear from Figure 3 that the Brønsted acid functionalized ILs with the Cl anion showed higher catalytic activity and gave a high LA yield compared to the ILs that have the NO_3_ and H_2_PO_4_ anions. The IL catalyst [C_4_SO_3_HPhim][Cl] showed 98% glucose conversion at 180 °C and 3 h reaction time with 63% yield of LA, while the catalyst [C_4_SO_3_HPhim][NO_3_] showed 91% glucose conversion with 56% LA yield. IL catalyst [C_4_SO_3_HPhim][ H_2_PO_4_] gave a maximum of 49% yield of the LA with 84% glucose conversion at 180 °C and 3 h reaction time. These catalyst activity trends of the IL catalysts confirmed that that the high acid strength is responsible for their high catalytic activity. The catalytic activities of the laboratory-synthesized Brønsted acidic ILs in LA synthesis from glucose can be arranged in the order of Cl^−^ > NO_3_^−^ > H_2_PO_4_^−^ which was found to be similar in their Hammett acidity order. However, the acidities of the Brønsted acidic ILs depend on the anions and affect the glucose conversion and yield of the LA.

#### 2.2.2. Effect of Reaction Temperature and Time on the LA Yield

Reaction temperature and time had significant effects on the LA yield. To optimize the reaction temperature for maximum product yield, the reactions were carried out at the temperature range of 140–200 °C. The effect of the reaction temperature on the LA yield from glucose conversion using Brønsted acidic IL as catalyst is shown in Figure 4a. From Figure 4a, it is clear that as the reaction temperature increased from 140 °C to 180 °C, the yield of LA increased significantly. At 180 °C, the highest yield of 63% LA was obtained from glucose. Upon further increases of the reaction temperature above 180 °C, no significant increase in the yield of LA was observed. At this temperature, the glucose conversion was almost 100%, and hence the yield of the LA was also at its maximum. At the higher reaction temperatures beyond 180 °C, glucose and intermediate products such as fructose and 5-HMF yielded a black insoluble humin side product catalyzed by the acidic reaction medium reducing the yield to the desired product LA. Therefore, 180 °C was selected as the optimum reaction temperature for glucose conversion. Similar to the temperature effect, the yield of the LA varied with the reaction time. The effect of the reaction time on the LA yield is shown in Figure 4b. At the initial stage of the reaction, the yield of LA is very low; at 1 h reaction time the yield of the LA was only 37% and reached 49% within 2 h of reaction time. As the reaction time increased up to 3 h, a maximum of 63 ± 2.0% yield of the LA was observed. Upon further increases in the reaction time up to 3.5 h, there was no significant change on the LA yield observed. Hence, 3 h is the optimum reaction time for glucose conversion to LA.

#### 2.2.3. Effect of the Ionic Liquid Amount

The effect of the amount of IL catalyst on the LA yield was investigated, and the results are shown in Figure 5a. From the Figure 5a it is clear that, when the reaction was performed in the absence of the catalyst there was less than 8% yield of LA observed, which confirmed that the glucose conversion into LA was a kinetically controlled catalytic reaction. To optimize the amount of the IL acid catalyst for high LA yields, the concentration of catalyst was varied from 0.25 g to 1.0 g by keeping the other parameters constant (100 mg glucose, 180 °C temperature, and 3 h reaction time). The results demonstrated that the yield of LA gradually increased from 48% to 63% with an increase of the amount of IL from 0.25 g to 0.50 g. Further increases in the amount of catalyst from 0.5 g to 1.0 g showed no significant increase in the LA yield, which may have been attributed to the presence of a high amount of the catalyst loading, and the formation of the side products called humin became dominant. Humin formation might be responsible for the condensation reaction of glucose, fructose, and 5-HMF. The amount of 0.50 g of the IL catalyst amount was found to be optimum to obtain the maximum yield of LA.

### 2.3. Effect of Solvents on LA Yield

The effect of different solvent media, 1-butyl-3-methylimidazolium chloride [BMIM][Cl], water, and THF were investigated with the [C_4_SO_3_HPhim][Cl] catalyst in order to explore their effect on the glucose conversion. The results are represented in Figure 6b. From the Figure 5b, it can be seen that the catalytic activity of IL catalyst was best in water compared to that in [BMIM][Cl] and THF reaction medium. In pure water, the conversion of glucose was almost 100%, and the yield of the LA was 63%, whereas in the case of the [BMIM][Cl] and THF, a maximum of 79% and 65% of the glucose was achieved with 17% and 40% yield of the LA, respectively. These results may have been due to the mass transfer resistances in the high viscosity [BMIM][Cl] IL and the low solubility of THF in glucose leading to the reduced glucose conversion in both these solvents. On the other hand, water, which is least viscous among the three, plays a crucial role in enhancing the reaction rate and also the rehydration of the intermediate product 5-HMF into LA. Thus, the order of solvents for improved LA yield in this reaction is water > THF > [BMIM][Cl].

### 2.4. Recycling of the Ionic Liquid Catalyst

The recyclability of the IL catalyst was done following the previous reported methods [21,46]. In the first run, the reaction mixture consisting of glucose, an optimum amount of the IL acid catalyst, and deionized water was taken in a 25 mL autoclave at 180 °C for 3 h reaction time. After the completion of the reaction, the insoluble material from the reaction mixture was filtered out from aqueous solution containing products and IL. The products were then extracted using ethyl acetate (organic phase) four times, and the IL remained in the aqueous phase. The UV–vis spectra of standard LA and the IL is represented in Figure 6. As shown in Figure 6, standard LA had strong absorption in the UV–vis range with a characteristic absorption at a wavelength of 266 nm, whereas for the ionic liquid, a characteristic broad band appeared at 244 nm, which can be attributed to the presence of a benzene ring within the ionic liquid. The UV spectrum of the purified LA matched with the standard LA spectrum, which confirmed that LA can be purified from the reaction mixture, which was also confirmed by the ^13^C and ^1^H NMR spectrum of the purified LA (Appendix A). The next reaction cycle was then conducted by adding a fresh glucose feed in the aqueous phase and so on up to four cycles. The results obtained from the reusability study of the catalyst is shown in the Figure 6c, and the figure shows that the PIL catalyst showed good catalytic activity with 59% product yield, even after the 4th cycle.

### 2.5. Comparison with Previous Study

The catalytic activity of the laboratory-synthesized 2-phenyl-2-imidazoline-based ionic liquid catalyst was compared for the glucose conversion to LA with the earlier reported studies. N.A.S. Ramli and co-workers synthesized LA using functionalized ionic liquid catalyst that had Bronsted and Lewis acidity and reported an 18% yield of the LA with a high amount (10 g) of catalyst loading [47], whereas W. Wei and co-workers used a combined mineral (H_3_PO_4_) and Lewis acid (CrCl_3_) catalyst for the glucose conversion to LA. This study reported a 54% yield of the LA at 170 °C and 4 h reaction time [48]. Fan Yang and co-workers used mixed-acid catalyst systems for glucose conversion to LA, which contained four Lewis acids (FeCl_3_, CrCl_3_, ZnCl_2_, and CuCl_2_) with three Brønsted acids (H_2_SO_4_, HCl, and H_3_PO_4_). The results obtained from this study confirmed that H_3_PO_4_ with CrCL_3_ showed better results and reported a 50% yield of the LA at 170 °C and 4.5 h reaction time [49]. However, the use of the mineral acid catalyst with metal chloride created some serious issues, such as reactor corrosion, and its waste neutralization created hazardous effluents [50]. In place of a mineral acid catalyst, the use of a solid acid catalyst for LA production can be a good idea. Using a solid acid catalyst, H. Qu and co-workers used a metal–organic framework containing a Brønsted and Lewis acid catalyst for glucose conversion to LA and reported a 58% yield of the LA at 150 °C and 9 h reaction time [24]. Aharon Gedanken’s research group developed a Ga salt of molybdophosphoric acid (GaHPMo) for LA production from glucose conversion and reported a 56% yield of the desired LA at 150 °C and 10 h reaction time [30]. However, the use of the solid acid catalyst for LA production required a long reaction time. In this study, the activity of the ionic liquid [C_4_SO_3_HPhim][Cl] catalyst for the glucose conversion to LA yield was found equal to or higher than most catalysts under the optimal reaction condition. Using [C_4_SO_3_HPhim][Cl] catalyst, a good (63%) yield of the LA was obtained at 180 °C and 3 h reaction time. In addition, the use of the IL catalyst was more environmentally friendly compared to the use of mineral acid catalyst.

## 3. Experimental Section

### 3.1. Materials and Methods

Glucose, fructose (99%), formic acid (FA) (99%), 5-HMF (98%), and LA (99%) were purchased from Fisher Scientific, Mumbai, India; 1,6-anhydro-β-D-glucose (99%) and 2-phenyl-2-imidazoline were purchased from TCI chemicals, Hydrabad, India; and 1,4-butanesultone (99%) was purchased from Alfa Aesar, Sanghai, China. Diethyl ether, ethyl acetate, tetrahydrofuran (THF), and HCl were purchased from Merck, Mumbai, India.

### 3.2. Synthesis of the Brønsted Acidic Ionic Liquid Catalyst

Synthesis of the Brønsted acidic IL catalyst was achieved following the reported literature with the slight modification [51,52]; the structure of the synthesized IL catalyst is represented in Figure 7. Typically, equimolar amounts of 2-phenyl-2-imidazoline (1.462 g, 0.01 mol) and 1,4 butanesultone (1.362 g, 0.01 mol) were dissolved in chloroform into a 100 mL round bottom flask. The desired reactant mixture was stirred at 40 °C for the required reaction time until completion of the reaction. After completion of the reaction, solvent was removed under reduced pressure to obtain the desired zwitterion. Finally, the zwitterion was washed with dichloromethane (DCM) or chloroform (3 × 5) to remove impurities and unreacted materials and dried for 24 h under vacuum at room temperature. The required Brønsted acidic IL catalyst was synthesized by the neutralization of the zwitterion using equimolar amounts of the corresponding acids under continuous stirring at room temperature. The synthesized Brønsted acidic IL catalyst was washed with diethyl ether repeatedly and dried under vacuum before use.

### 3.3. Characterization of the Synthesized Brønsted Acidic Ionic Liquid Catalyst

The laboratory-synthesized IL catalyst was characterized using the ^1^H NMR, FT-IR, and thermal gravimetric analysis (TGA). The ^1^H NMR of the compounds was characterized using a Broker Spectrospin 400 MHZ instrument (Bruker Spectrospin, New Delhi, India). The ATR-FTIR transmission spectra of the laboratory-synthesized IL catalyst were recorded using a Nicolet 6700 FT-IR spectrometer equipped with a DTGS detector (Thermo Fisher Scientific, New Delhi, India). The FT-IR spectra of the catalyst was collected at room temperature (30 °C) in the wavenumber range of 400–4000 cm^−1^.

Thermogravimetric analysis (TGA) was performed to analyze the stability of the catalyst at higher temperatures using a TGA-Q 500 instrument (Perkin Elmer, New Delhi, India) with a nitrogen flow rate of 40 mL/min from 30 °C to 550 °C at a heating rate of 10 °C/min.

### 3.4. Measurement of the Brønsted Acidic Strength of the Ionic Liquids

The acidic strength of the laboratory-synthesized IL catalysts having different anions was determined using a previously reported method [53]. The required molar amount of 4-nitroaniline and 25 mM of the IL catalysts were dissolved in water and magnetically stirred for the required time. After the IL completely mixed with 4-nitroaniline, spectral scanning was processed from 300 nm to 550 nm by UV–Vis using a double beam Dynamica spectrophotometer (Model: Halo DB-20, Dynamica Scientific Ltd., New Delhi, India). The absorption spectrum was compared to the maximum absorption wavelength of 4-nitroaniline (at 380–381 nm). Finally, the acidity of the synthesized catalysts was calculated using the Hammett equation.
(1)$$H_{0}=\mathrm{pK}\;{\left(I\right)}_{\mathrm{aq}}+\log\left(\frac{\left[I\right]}{\left[{\mathrm{IH}}^{+}\right]}\right)$$
where [I] represents the molar concentration of unprotonated 4-nitroaniline, [IH^+^] is the molar concentration of protonated forms of the 4-nitroaniline in water, and pK(I)_aq_ is the pKa value of the 4-nitroaniline. The values of the Hammett acidity function ($H_{0}$) of the IL catalysts are shown in Table 1.

### 3.5. General Procedure for Experimental Catalytic Conversion of Glucose to Levulinic Acid

All of the experiments involved in the glucose conversion to LA were performed in a 25 mL batch autoclave. The reactor was loaded with the required amount of deionized water and 100 mg of glucose along with the required amount of catalyst. The reactor was then sealed and placed in a silicone oil bath that was preheated to the required reaction temperature on a Tarson hot plate magnetic stirrer. The desired reaction mixture was stirred with controlled stirring speed over fixed time intervals. After the completion of the reaction, the reactor was quenched in cold water to stop undesired reactions. The sample was collected from the autoclave and filtered using a syringe filter. The liquid samples were analyzed using high pressure liquid chromatography (HPLC). Quantitative analysis of the reaction products was calculated following a standard calibration curve drawn from commercially available standard samples.

### 3.6. Product Analysis

The liquid products were quantified by high pressure liquid chromatography (HPLC) equipped with HPX-87H column. The samples were filtered using a 0.22 µm syringe filter prior to analysis. During this process, 5 mM sulphuric acid was used as the mobile phase with a constant flow rate of 0.5 mL/min equipped with an RID detector. The column and detector temperatures were kept constant at 65 °C and 35 °C, respectively. Quantification of the product was done based on the standard curve obtained by using an external standard.
(2)$$\mathrm{Reactant}\;\mathrm{conversion}\;(\%)\;=\;\frac{\mathrm{Initial}\;\mathrm{no}.\;\mathrm{of}\;\mathrm{moles}\;\mathrm{of}\;\mathrm{the}\;\mathrm{glucose}\;-\;\mathrm{remaining}\;\mathrm{no}.\;\mathrm{of}\;\mathrm{the}\;\mathrm{moles}\;\mathrm{of}\;\mathrm{glucose}\;}{\mathrm{Initial}\;\mathrm{no}.\;\mathrm{of}\;\mathrm{moles}\;\mathrm{of}\;\mathrm{glucsoe}}\times 100$$
(3)$$\mathrm{Product}\;\mathrm{yield}\;(\%)\;=\;\frac{\mathrm{Moles}\;\mathrm{of}\;\mathrm{LA}\;\mathrm{obtained}\;\mathrm{from}\;\mathrm{the}\;\mathrm{reaction}}{\mathrm{Initial}\;\mathrm{moles}\;\mathrm{of}\;\mathrm{the}\;\mathrm{glucose}}\times 100$$

### 3.7. Reusability of the Ionic Liquid Catalyst

The reaction mixture after the catalytic reaction was passed through the 0.22 µm syringe filter to remove any insoluble solid product. Ethyl acetate was selected as an extracting organic solvent that can easily extract LA from the aqueous phase, since the IL is insoluble in ethyl acetate. Through facile phase separation, the aqueous and organic layer were separated. From the organic phase, the LA was collected by separating the organic phase using a rotary evaporator under reduced pressure. The pure isolated product was then characterized by ^1^H and ^13^C NMR. Fresh glucose reactant was then added to the aqueous layer to study the reusability of the catalyst for the next four consecutive reactions.

## 4. Conclusions

In this study, Brønsted acid functionalized 2-phenyl-2-imidazoline-based IL catalysts with different anions, namely [C_4_SO_3_HPhim][Cl], [C_4_SO_3_HPhim][NO_3_], and [C_4_SO_3_HPhim][H_2_PO_4_] were successfully synthesized in the laboratory and characterized using ^1^H NMR, FT-IR, and TGA spectroscopic techniques. The spectroscopic characterization of the catalyst confirmed the successful synthesis of the catalyst. From the TGA characterization, it was seen that the catalysts have good thermal stability up to 200 °C, which means they are very stable at the optimum reaction temperature of the glucose conversion to LA, which is below 200 °C. The Brønsted acidity trend of these laboratory-synthesized IL catalysts was experimentally examined using UV–vis spectroscopy and found to follow the order [C_4_SO_3_HPhim][Cl] > [C_4_SO_3_HPhim][NO_3_] > [C_4_SO_3_HPhim][H_2_PO_4_], which is also the activity trend of the catalyst in LA synthesis from glucose. The catalytic activity of the synthesized catalyst was tested in the glucose conversion to LA in a batch autoclave with an optimum amount of catalyst (0.5 g) using water as a solvent, and a maximum yield of 63% LA was achieved with 99% glucose conversion at 180 °C reaction temperature and 3 h reaction time. The IL catalyst [C_4_SO_3_HPhim][Cl] was found to be reusable up to four times with no significant loss of LA yield. To conclude, this research promises a benign ionic liquid catalyst and solvent media for maximum glucose conversion of the very versatile and important lignocellulosic biomass-derived platform chemical levulinic acid.

## Figures and Tables

**Figure 1 molecules-26-00348-f001:**
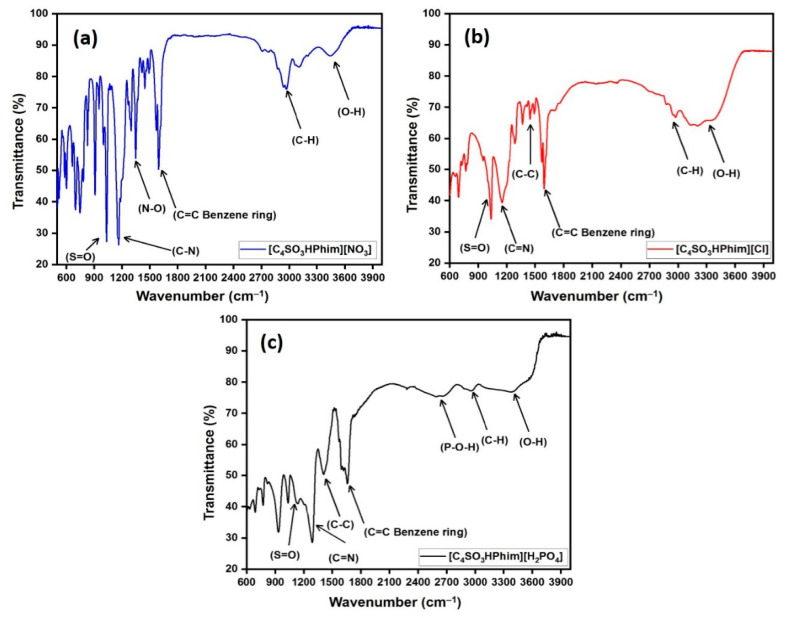
FT-IR spectrum of the synthesized ILs (**a**–**c**).

**Figure 2 molecules-26-00348-f002:**
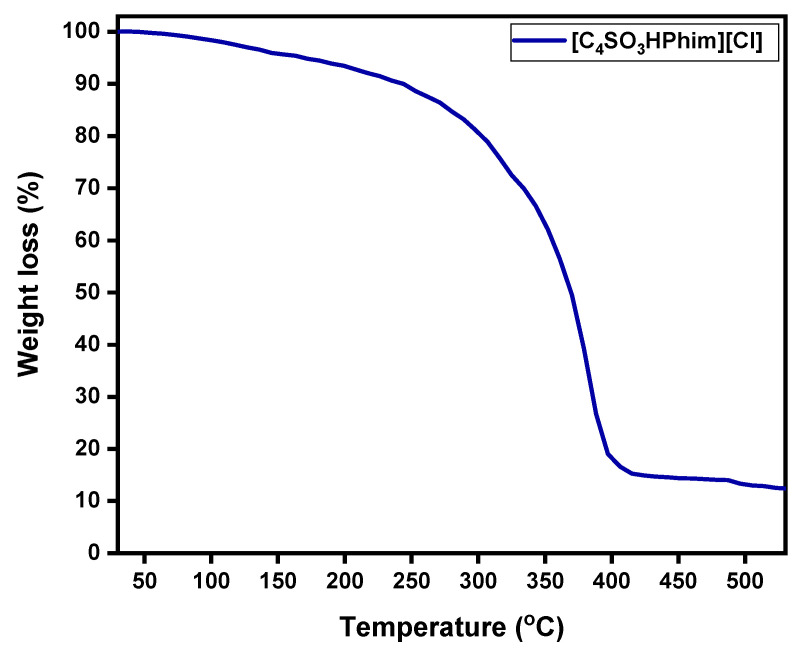
TGA of the synthesized ionic liquid catalyst ([C_4_SO_3_HPhim][Cl]).

**Figure 3 molecules-26-00348-f003:**
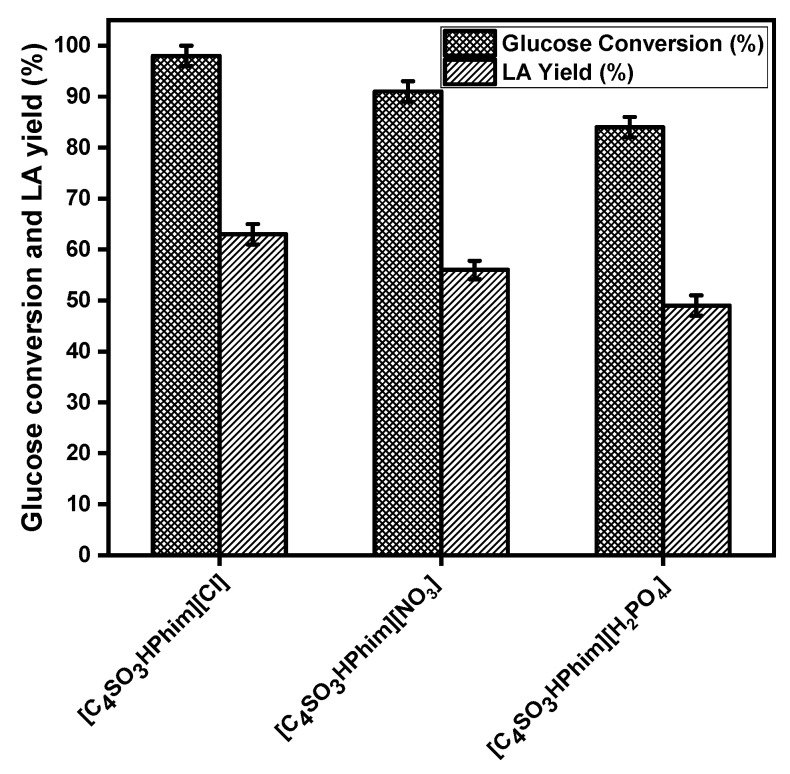
Effect of the Brønsted acidic ionic liquids on the conversion of glucose to LA. Reaction conditions: ionic liquid = 500 mg, 100 mg glucose, 2.0 mL water, temp = 180 °C, t = 3 h.

**Figure 4 molecules-26-00348-f004:**
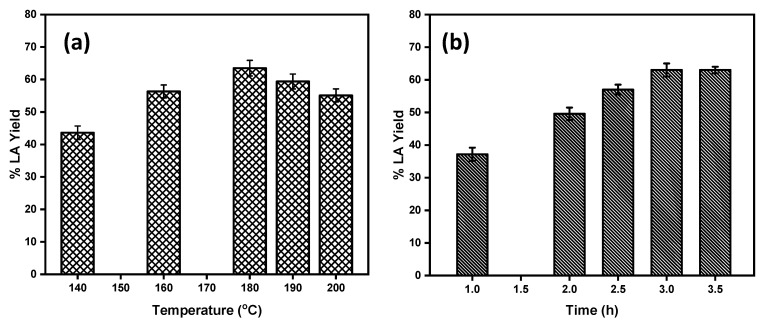
(**a**) Effect of the reaction temperature on the product yield. (**b**) Effect of the reaction time on LA yield. Reaction conditions: 500 mg [C_4_SO_3_HPhim][Cl], 100 mg glucose, 2.0 mL of water.

**Figure 5 molecules-26-00348-f005:**
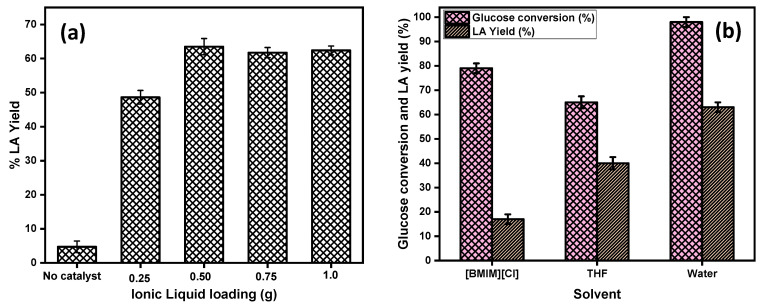
(**a**) Effect of the ionic liquid loading on the product yield; reaction conditions: 100 mg glucose, 2.0 mL of water, temp = 180 °C. (**b**) Effect of the solvents on the glucose conversion and LA yield; reaction conditions: 100 mg glucose, 2.0 mL solvent, t = 3 h, temp = 180 °C, catalyst [C_4_SO_3_HPhim][Cl] = 500 mg.

**Figure 6 molecules-26-00348-f006:**
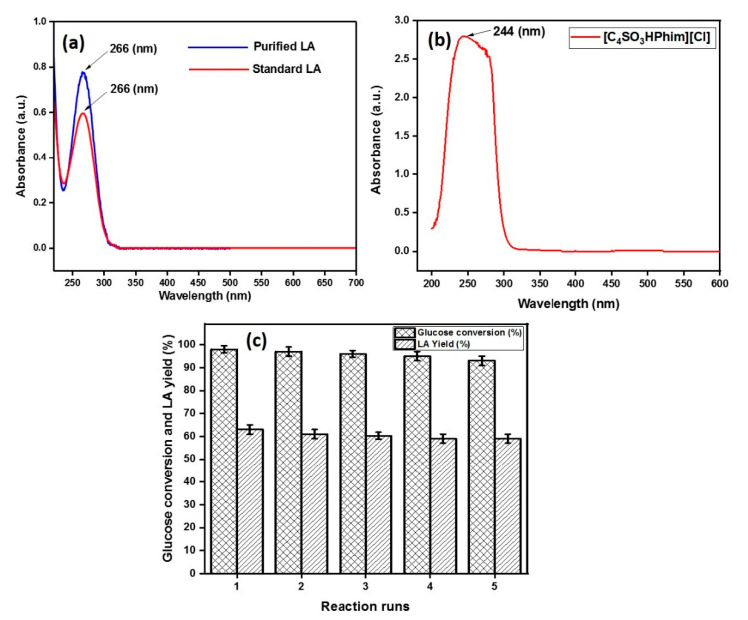
UV–vis spectrum of purified LA and standard LA (**a**) UV–vis spectrum of the IL catalyst [C_4_SO_3_HPhim][Cl] (**b**) Reusability of the IL catalyst [C_4_SO_3_HPhim][Cl] (**c**).

**Figure 7 molecules-26-00348-f007:**
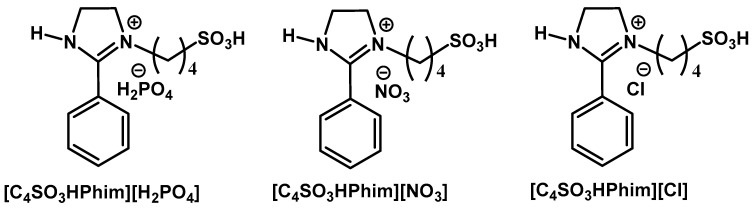
Structure of the ionic liquid catalyst used in this study for glucose conversion to levulinic acid (LA).

**Table 1 molecules-26-00348-t001:** The values of Hammett acidity function ($H_{0}$) for all Brønsted acidic IL catalysts.

ILs	A_max_	[I](%)	[IH^+^]%	*H* _0_
**None**	1	100	0	-
**[C_4_SO_3_HPhim][NO_3_]**	0.79	98.13	01.86	2.51
**[C_4_SO_3_HPhim][Cl]**	0.77	96.14	03.85	2.17
**[C_4_SO_3_HPhim][H_2_PO_4_]**	0.80	99.37	0.62	3.00

## Data Availability

The data presented in this work are available on request from the corresponding author.

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
