# Peer review of "Catalytic Conversion of Glucose into Levulinic Acid Using 2-Phenyl-2-Imidazoline Based Ionic Liquid Catalyst"

_molecules, 2021, doi:10.3390/molecules26020348_

Round 1
Reviewer 1 Report
The work by Upadhyayula and co-workers reported the catalytic conversion of glucose into levulinic acid using 2-phenyl-2-imidazoline based ionic liquids. Corresponding ionic liquids were synthesized by a two-step method, which were then characterized and used for glucose conversion. Effect of the anion structure on the catalytic activity was revealed, and it was found that the ionic liquid with stronger acidity exhibited higher catalytic performance. This study expands the ionic liquid family for biomass conversion. After addressing the following questions/issues, this paper could be published in Molecules.
- Introduction: Specific values should be given in the statement of “…but many of these heterogeneous catalysts give low selective yield of the desired products and require higher catalyst/substrate ratios, high reaction temperature and long reaction time”. For instance, what were the reaction temperature and time in the reported process? The reaction temperature used in this study (i.e., 180 °C) seemed to be higher than those reported by others (Table 2).
- The only difference of these three synthesized ionic liquids was the anion. However, the FT-IR spectra of these ionic liquids seemed to be quite different. The authors should analyze the results in detail.
- The ionic liquids were analyzed by TGA as mentioned in the Experimental and Conclusions sections. But no TGA data was mentioned in the Results and Discussion section.
- Recyclability: although the ionic liquid catalyst and ethyl acetate are immiscible with each other, a certain amount of the ionic liquid could be dissolved in the ethyl acetate phase. The authors should detect whether there was any ionic liquid left in the product using other techniques like mass spectrometry, UV (analysis by NMR was not enough). If the ionic liquid catalyst could be distributed into the product, it must affect the recyclability as the amount of catalyst decreased. The conversion of glucose in each cycle should be given in Figure 6c.
- What possible side reactions could occur in the conversion of glucose into LA? Why the increased ionic liquid catalyst dosage (e.g., 0.75 g or 1.0 g) led to the formation of side products?
- English expression of the paper should be improved, e.g., “has attracted a great attention” should be “has attracted great attention”; there was grammar mistake in “…ionic liquids (ILs) are one such good choice that can be used as catalysts and solvents conversion of biomass to products and intermediates like LA”, etc.
Author Response
Reviewer 1: Comments and Suggestions for Authors
The work by Upadhyayula and co-workers reported the catalytic conversion of glucose into levulinic acid using 2-phenyl-2-imidazoline based ionic liquids. Corresponding ionic liquids were synthesized by a two-step method, which were then characterized and used for glucose conversion. Effect of the anion structure on the catalytic activity was revealed, and it was found that the ionic liquid with stronger acidity exhibited higher catalytic performance. This study expands the ionic liquid family for biomass conversion. After addressing the following questions/issues, this paper could be published in Molecules.
- Introduction: Specific values should be given in the statement of “…but many of these heterogeneous catalysts give low selective yield of the desired products and require higher catalyst/substrate ratios, high reaction temperature and long reaction time”. For instance, what were the reaction temperature and time in the reported process? The reaction temperature used in this study (i.e., 180 °C) seemed to be higher than those reported by others (Table 2).
Response: The authors thank the reviewer for his/ her valuable observations and suggestions. The above suggestions have been incorporated in the revised manuscript in line number 60-78. “Many of these problems can be avoided by replacing the liquid catalysts by solid acid catalysts. However, these heterogeneous solid catalysts gave low selective yield of the desired product and required either higher catalyst to substrate ratios and/ or long (more than 3 hours) reaction times [28,29]. Haonan Qu and co-workers used metal-organic frameworks which contain Brønsted and Lewis acidity as solid catalysts and reported 57.9% yield of LA at 150 °C after 9 h reaction time [30]. Vijay Bhooshan Kumar and co-workers used Ga salt of molybdophosphoric acid (GaHPMo) for glucose conversion to LA and reported 56% yield at 150 °C and 10 h reaction time [31]. Wei Zeng and co-workers used MFI-Type Zeolite (ZRP) and reported 35% LA yield from glucose conversion at 180 °C and 8h reaction time [32]. In this context, ionic liquids (ILs) are one such good choice that can be used as catalysts and solvents for the biomass conversion into value-added products and intermediates like LA [33].”
“Yue Shen et al., attempted cellulose conversion to LA and obtained a highest yield of 39.4% after 120 min using lab synthesized 1-(4-sulfonic acid) butyl-3-methylimidazolium hydrogen sulphate ([BSMim]HSO4) with addition of H2O [39].”
- The only difference of these three synthesized ionic liquids was the anion. However, the FT-IR spectra of these ionic liquids seemed to be quite different. The authors should analyse the results in detail.
Response: The FT-IR spectrum of the characterized IL catalysts with their characteristics peaks is shown in the Figure 2(a-c) after detailed analysis as suggested by the reviewer.
- The ionic liquids were analyzed by TGA as mentioned in the Experimental and Conclusions sections. But no TGA data was mentioned in the Results and Discussion section.
Response: TGA data for the IL catalyst [C4SO3HPhim][Cl] has now been incorporated in the revised manuscript as Figure 3 as suggested by the reviewer.
- Recyclability: although the ionic liquid catalyst and ethyl acetate are immiscible with each other, a certain amount of the ionic liquid could be dissolved in the ethyl acetate phase. The authors should detect whether there was any ionic liquid left in the product using other techniques like mass spectrometry, UV (analysis by NMR was not enough). If the ionic liquid catalyst could be distributed into the product, it must affect the recyclability as the amount of catalyst decreased. The conversion of glucose in each cycle should be given in Figure 6c.
Response: The reusability of the IL catalyst was tested following reported methods (Ind. Eng. Chem. Res., 2016, 55(42), 11044–11051, Bioresour. Technol., 2015, 192, 812-816, J. Mol. Catal. A: Chem., 2012, 357, 11-18, New J. Chem., 2018, 42, 228-236). As suggested by the reviewer, UV-vis., 1H and 13C NMR spectrum of the purified LA from glucose conversion has been included in the revised manuscript in the Figure 7 (a-d). From the UV-vis, 1H and 13C NMR spectrum, it can be seen that the purified product doesn’t contain any trace of the IL in the product confirming maximum amount of the IL catalyst recovery and recyclability of the IL catalyst.
- What possible side reactions could occur in the conversion of glucose into LA? Why the increased ionic liquid catalyst dosage (e.g., 0.75 g or 1.0 g) led to the formation of side products?
Response: The side reaction is the formation of a black charry material called humin as the byproduct which is detected during the catalytic conversion of the glucose to LA. The formation of humin is due to the self or cross polymerization of glucose or fructose and/ or 5-HMF (intermediate product participating in polymerization side reaction) catalyzed by acid (Green Chem., 18 (2016), 1983-1993, Green Chem. 2011, 13(7), 1676−1679, Chemical Engineering Journal 307 (2017) 389–398). The acid concentration of the overall reaction medium increases as the amount of the ionic liquid catalyst is increased which results in increased self-polymerization reaction and promotes the humin formation.
- English expression of the paper should be improved, e.g., “has attracted a great attention” should be “has attracted great attention”; there was grammar mistake in “…ionic liquids (ILs) are one such good choice that can be used as catalysts and solvents conversion of biomass to products and intermediates like LA”, etc.
Response: Authors thank the reviewer for his/ her valuable suggestion. The whole manuscript has been revised with English grammar and expression corrections in line numbers 12-18, 19-21, 38-43, 47-50, 63-66.
Reviewer 2 Report
The submitted work is an interesting contribution to the general knowledge about the production of levulinic acid using ionic liquids. This work can be accepted for publication after addressing some issues listed below.
- In the introduction section stated that from glucose several types of pivot chemicals can be obtained. One of them is 5-HMF but author stated a specific work. It would be better to cite more comprehensive review, e.g. of Zakrzewska et al. Chem. Rev. 2011, 111, 397. Additionally, when stated about other biomass derived solvents, the most recent review on this should be mentioned, Kochepka et al. Acta Innovations, 2020, 35, 29.
- Along the entire work when authors state about the yield of conversion it is unclear if % is given in mass or mole%.
- The comparison with previous work is very scarce. Instead of giving a table authors should discuss the pros and cons of the method in light of the obtained results and compare them to literature data.
After addressing the aforementioned aspects, this work can be accepted for publication.
Author Response
Reviewer 2: Comments and Suggestions for Authors
The submitted work is an interesting contribution to the general knowledge about the production of levulinic acid using ionic liquids. This work can be accepted for publication after addressing some issues listed below.
- In the introduction section stated that from glucose several types of pivot chemicals can be obtained. One of them is 5-HMF but author stated a specific work. It would be better to cite more comprehensive review, e.g. of Zakrzewska et al. Chem. Rev. 2011, 111, 397. Additionally, when stated about other biomass derived solvents, the most recent review on this should be mentioned, Kochepka et al. Acta Innovations, 2020, 35, 29.
Response: The authors thank the reviewer for his/ her valuable observations. The suggested valuable references have been cited at Reference numbers 10 and 19 in the revised manuscript.
- Along the entire work when authors state about the yield of conversion it is unclear if % is given in mass or mole%.
Response: The conversion and yield of the product is calculated in mole % and this is now incorporated in the revised manuscript in line numbers 152 and 153.
- The comparison with previous work is very scarce. Instead of giving a table authors should discuss the pros and cons of the method in light of the obtained results and compare them to literature data.
Response: The Table 2 has been removed in the revised manuscript and a comparison of the present results with that of previous reports is given discussing with pros and cons of the method in the light of the obtained results in line numbers 305-328 in the revised manuscript.
After addressing the aforementioned aspects, this work can be accepted for publication.
Response: Author thank the reviewer for his/her valuable recommendations.
Round 2
Reviewer 1 Report
The manuscript has been improved a lot after the authors made the corrections. However, some modifications are still needed.
- Abstract: The two statements: “Among these, 1-butyl sulfonic acid-2-phenylimodazoline chloride [C4SO3HPhim][Cl] demonstrated good catalytic activity in the glucose conversion to levulinic acid as compared to the [C4SO3HPhim][NO3] and [C4SO3HPhim][H2PO4] IL catalysts.”, “The activity of these Brønsted acidic ionic liquid catalyst were found to follow the order: [C4SO3HPhim][Cl] 17 >[C4SO3HPhim][NO3]>[C4SO3HPhim][H2PO4].” have similar meanings.
- The 5% onset decomposition temperature of the ionic liquid should be given in Section 3.1.3.
- Caption for Figure 3 is incorrect.
- The UV-vis spectra of standard LA and the IL should also be included in Figure 7c.
Reviewer 2 Report
The authors performed suggested changes and significantly imporved the submitted work. The only minor issue are errors in the reference list as well as incoherence with inclusion of DOI in some references and not all.
Author Response
Comments and Suggestions for Authors by Reviewer 2:
The authors performed suggested changes and significantly improved the submitted work. The only minor issue are errors in the reference list as well as incoherence with inclusion of DOI in some references and not all.
Response: The error has now been corrected in the reference list with inclusion of DOI in all in the revised manuscript.